# Mainstreaming Nature-Based Solutions in City Planning: Examining Scale, Focus, and Visibility as Drivers of Intervention Success in Liverpool, UK

Ian Mell [1,*], Sarah Clement [2] and Fearghus O'Sullivan [1]

1   Department of Planning & Environmental Management, University of Manchester, Manchester M13 9PL, UK
2   College of Science, Australian National University, Canberra, ACT 2601, Australia
*   Correspondence: ian.mell@manchester.ac.uk

**Abstract:** Nature-based solutions (NBS) have been central to the European Union's drive to address climate change, ecological degradation, and promote urban prosperity. Via an examination of the Horizon 2020-funded URBAN GreenUP project in Liverpool, this paper explores mainstreaming NBS in city planning. It uses evidence from pre- and post-intervention surveys with Liverpool residents and interviews with local business, environmental, government, and community sector experts to illustrate how a complex interplay of scale, location, focus, and visibility of NBS influences perceptions of the added value of NBS. This paper highlights the requirement that NBS interventions be bespoke and responsive to the overarching needs of residents and other stakeholders. Moreover, we underscore the importance of expert input into the design, location, and maintenance of NBS and call for these key drivers of successful delivery to be better integrated into work programs. This paper also notes that the type and size of NBS interventions impact perceptions of their value, with smaller projects being viewed as less socially and ecologically valuable compared to larger investments. We conclude that while small-scale NBS can support climatic, health, or ecological improvements in specific instances, strategic, larger-scale, and more visible investments are required to accrue substantive benefits and gain acceptance of NBS as a legitimate and effective planning tool.

**Keywords:** Nature-Based Solutions; urban planning; community perceptions; urban nature; biodiversity; multi-functionality; climate change; green infrastructure

## 1. Introduction

The rise of Nature-based solutions (hereafter NBSs) has been catalyzed by significant investment from the European Union (EU) through its Research and Innovation (R&I) portfolio. EU funding aims to build the evidence base for how nature-focused interventions can address climate change, improve public health and well-being, support economic growth, and promote urban renewal [1]. The Horizon 2020 schemes focusing on NBSs fund research and demonstration projects to showcase the breadth of opportunities available to planners, politicians, the environment, and the public sectors to solve public problems and integrate more ecologically sustainable development into urban development [2]. Through a broad program of micro/singular, street, and area/neighborhood-based interventions, NBSs have been implemented in European cities to test the positive impact that nature-focused interventions can have at multiple scales (micro, e.g., a lamp post; site, e.g., a park or building); street; neighborhood; and across different urban contexts [3]. The following uses "scale" to define the size of a NBS intervention. In addition, the focus of NBS is reflective of their multiple socio-economic and ecological functions and how they aid the delivery of climate change adaptation/mitigation, health and well-being, economic prosperity, and improved quality of life. At the same time, the visibility of NBS relates to the ease with which NBS interventions are seen and interacted with in an urban context. The goal is ultimately transformative, i.e., to provide ecological and socio-economic evidence

for NBS, enhance their visibility to potential users, and mainstream them within urban planning and regeneration design and delivery.

To examine the added value of NBS in delivering ecological and socio-economic benefits in cities, we leveraged insights from a six-year project to design, implement, and test the contribution of nature-centric projects to solving urban challenges [4]. The paper uses the Horizon 2020-funded URBAN GreenUP project as a case study, and specifically the interventions delivered in Liverpool (UK), to illustrate the complexity of translating the theoretical promise of NBS into the practice of greening highly urbanized environments. The paper sets out to answer the following:

1.  Which NBS are considered most appropriate by residents, businesses, and other communities of interest to address a range of sociocultural, economic, and ecological challenges in the Liverpool case study area?
2.  What barriers can hinder the delivery of NBS within a high-density urban area?
3.  What are the most appropriate NBS options in terms of scale (micro, site, street, neighborhood), location (urban, urban-fringe, rural), NBS ecological and socio-economic function, visibility, and interactivity that can be used to address the widest range of issues impacting high-density urban areas?

The paper presents two sets of interlinked evidence to answer these questions: (1) a survey of residents in Liverpool examining the perception of existing GI and URBAN GreenUP-funded NBS; and (2) insights from development, third, and environmental organizations. This directly responds to the enthusiastic advocacy for urban NBS in policy and practice, providing evidence from a live demonstration project that tests the promise of NBS in practice. This evidence bridges professionals, e.g., local government and representatives of environmental organizations, and local perspectives on the perceived added ecological and/or socio-economic value provided by investment in NBS. This allows for a detailed commentary on local and strategic considerations for implementing NBS in Liverpool. This is of global interest, considering it was a centerpiece of EU investment and part of a wider effort to leverage the power of NBS in cities as other cities will face similar challenges. Within this context, successful delivery is framed as an investment that local participants perceive as positively enhancing the quality of quality of life socially, economically, or ecologically. Success is not presented as calculable (or quantitatively evidenced) improvements in urban ecosystem functionality, as this is beyond the scope of this paper. Therefore, the analysis presented relates to the perceptions of business, environmental, and residential stakeholders of the additional benefits that NBS provides in Liverpool. The paper showcases where links between climate adaptation, improved access to nature, enhanced air quality, and improved health and well-being can be enhanced through NBS investment. However, our research also underscores that communication of the benefits of investment in NBS, co-design with residents, the third sector, and environmental organizations, and delivery that meets identified local needs are core factors that must be foregrounded in all cities to guide investment.

Overall, the paper argues for a locally grounded appreciation of which NBS may be appropriate in urban areas and consideration of what benefits and functions are needed to maximize the value of such interventions for resolving complex challenges. While NBS has been automatically accepted as a public good, we call for consideration in the proposal and design stages of what socio-economic and/or ecological "additionality" any NBS intervention will provide and to whom [5]. This supports the evolving discussions of NBS interventions developed by Kabisch, Frantzeskaki, and Hansen [6], who, as part of the wider discourse supporting NBS, suggest that effective NBS investment should be structured against five core principles: (a) systematic understanding; (b) benefits to people and biodiversity; (c) inclusive solutions that are long-term; (d) context consideration; and (e) communication and learning. While the debates presented in this paper acknowledge the value of such framings, the discussion presented in the case of Liverpool placed an increased emphasis on considerations of NBS in terms of (i) elements, (ii) functions,

(iii) benefits, and (iv) beneficiaries to ensure that locally appropriate investments are delivered.

## 2. Framing NBS in Urban Planning

NBS emerged internationally as an approach to resolving linked challenges relating to climate change, biodiversity loss, and community livelihoods [7]. Although the concept originally focused on the conservation, restoration, and enhancement of natural ecosystems and broader landscapes, the focus of these debates in Europe shifted to greening cities, where the majority of people live and where environmental challenges are most acute [8]. Approximately 70% of the EU's population lives in urban areas, driving significant changes in the functionality of linked socio-economic and ecological systems. This includes impacts on water quality and quantity, biodiversity loss, and degraded air quality, with consequences for ecological function, human health, and the economic viability of cities. The Horizon 2020 program aims to add to the evidence base to understand how investment in "nature" in its myriad forms (e.g., street trees, green facades, parks, or sustainable drainage) can act as a viable solution to the problems associated with growth and unsustainable development patterns in cities.

By building on conversations about the benefits of nature, e.g., ecosystem services [9] and the connective [10], accessible, and multi-functional principles of green infrastructure (GI) [11,12], NBSs are being promoted as an innovative way to enhance these benefits by integrating ecological thinking into engineered, i.e., built environment, systems [13]. NBS as a term is new, but conceptually, it is built on decades of research discussing the value of an investment in urban nature, for example, in the urban ecology literature [14,15]. However, the promotion of "nature" as the central principle of investment does differ from previous forms of green space and landscape development [16,17]. In such a scenario, GI, or greenspace, could be positioned as an overarching concept, putting a conceptual and thematic structure in place that contextualizes investment in nature as essential infrastructure [18]. Practice-based delivery can subsequently promote using NBS as the action component of a wider, environment-centric framework to deliver ecologically focused development. By working with nature as a means to deliver on the objectives of core goals rather than as an afterthought, investments in NBS can offer cost-effective and responsive forms of urban management that support greener and more sustainable growth in cities [19,20].

The capacity of government decision-makers and the environmental and business sectors to implement NBS varies substantially geographically and across governance levels (e.g., local, regional, and national) [21]. Consequently, although advocates in environmental organizations and academia have argued convincingly for investment in NBS, there remain diverse views of the added value that NBS can provide [22]. Current debates on NBS are starting to address this issue to ensure that the technical, legal, and political challenges faced by practitioners, scientists, and decision-makers working in cognate sustainability disciplines are more effectively integrated into urban development practice. Consequently, despite the relative infancy of NBS as an academic subject, it has emerged as core terminology within urban nature debates in Europe. This shift is visible in the catalogs of NBS typologies and investment options being proposed in the literature. These include the use of urban forestry, sustainable drainage, increased biodiverse planting, green wall/roof/facade technology, parklets and parks, urban agriculture, and roadside verge pollinator enhancements [1,18,23]. More technological solutions are also being debated, utilizing sensors to examine heat, pollution, water, and biodiversity change linked to variations in the form and function of each of the types of NBS noted above. The breadth of interventions available highlights the need for an understanding of NBS to be informed by examples of their value in practice.

The benefits of ecologically centered investment include the delivery of comparable functionality to engineered solutions but at lower costs and more responsiveness to changes in the fabric of an urban environment [24]; thus, they offer a 'dynamic mutability' to the pressures placed on urban landscapes [19]. Therefore, taking a wide perspective on what

can be considered NBS is a useful approach to addressing such variability. Furthermore, the research literature suggests that NBS optimize the benefits of ecological systems within the built environment, promoting a more nuanced appreciation of "nature" within praxis to ensure different stakeholders can more effectively engage with NBS compared to traditional engineered solutions. When aligned with a consideration of NBS elements, functions, benefits, and beneficiaries, it is possible to apply a more holistic framing to NBS, examining what is delivered, what change they are promoting, and how local people can engage with and benefit from them [18].

## 3. The URBAN GreenUP Project and Greenspace Planning in Liverpool

These assertions can be tested by examining the design, delivery, and monitoring of NBS interventions associated with the URBAN GreenUP project. URBAN GreenUP (https://www.urbangreenup.eu/, accessed on 7 July 2023) is a 5-year EU-funded R&I project testing the value of integrating innovative NBS in urban areas. Due to delays in the delivery of NBS caused by COVID-19, URBAN GreenUP was extended by 12 months to become a 6-year program. The project comprised a consortium of 27 universities and research institutions, Small–Medium Enterprises (SMEs), local governments, and environmental organizations located in eight countries (China, Columbia, Germany, Italy, Spain, Turkey, the UK, and Vietnam).

Liverpool (UK) is one of three front-runner cities, along with Izmir (Turkey) and Valladolid (Spain), that led the delivery of NBS interventions and the development of a transferable methodology for planning and implementing NBS. The project also involved five follower cities (Chengdu in China, Ludwigsburg in Germany, Mantova in Italy, Medellin in Columbia, and Quy Nhon in Vietnam), which tested the project's innovations and methods. Each partner city brings a wealth of local government, environmental, and technological expertise to the project that has been integrated to aid in the identification of solutions to a range of sociocultural, economic, and ecological problems. The three front-runner cities developed portfolios of NBS interventions, which have subsequently been expanded into a series of renaturing strategies for cities. The project also facilitated a more holistic understanding of the complexity associated with NBS delivery, as it has worked across varied geographic, climatic, political, and governance systems.

The URBAN GreenUP project does not sit in isolation but is one of three EU Horizon 2020-funded NBS R&I projects. Its sister projects, Connecting Nature (https://connectingnature.eu/, accessed on 7 July 2023) and Grow Green (https://growgreenproject.eu/, accessed on 7 July 2023), are comparable to URBAN GreenUP in terms of their consortium composition. Variations are visible in each project's strategic objectives. Grow Green in Manchester focuses on a single large neighborhood-scale project, West Gorton Sponge Park, and the Glasgow (UK)-based components of Connecting Nature, including a range of small- and medium-sized interventions. The breadth of delivery has provided important insights for the EU, enabling them to examine what works, how it works, and where barriers remain to successful NBS interventions.

While NBS should be delivered at a landscape scale to meet global standards [25], retrofitting NBS within existing 'hard' urban forms makes this challenging. This challenge is evident in the portfolio of NBS interventions delivered by URBAN GreenUP, which were at a site and street scale. Figure 1a–d illustrates examples of the interventions, which include green walls, pollinator lamp posts, biodiverse pollinator street planting, and investment in street trees to add shade, promote pollination, and intercept pollution and rainfall. In addition, Table 1 provides a profile of each research/investment area and the NBS interventions located in each. Moreover, ecological floating islands, deculverting/bioretention and sustainable drainage works, and urban garden bio-filters were all delivered to examine the potential impact that NBS could have on people, the environment, and the local economy. A full portfolio of investments can be accessed via the project webpages: https://www.urbangreenup.eu/solutions/, accessed on 7 July 2023.

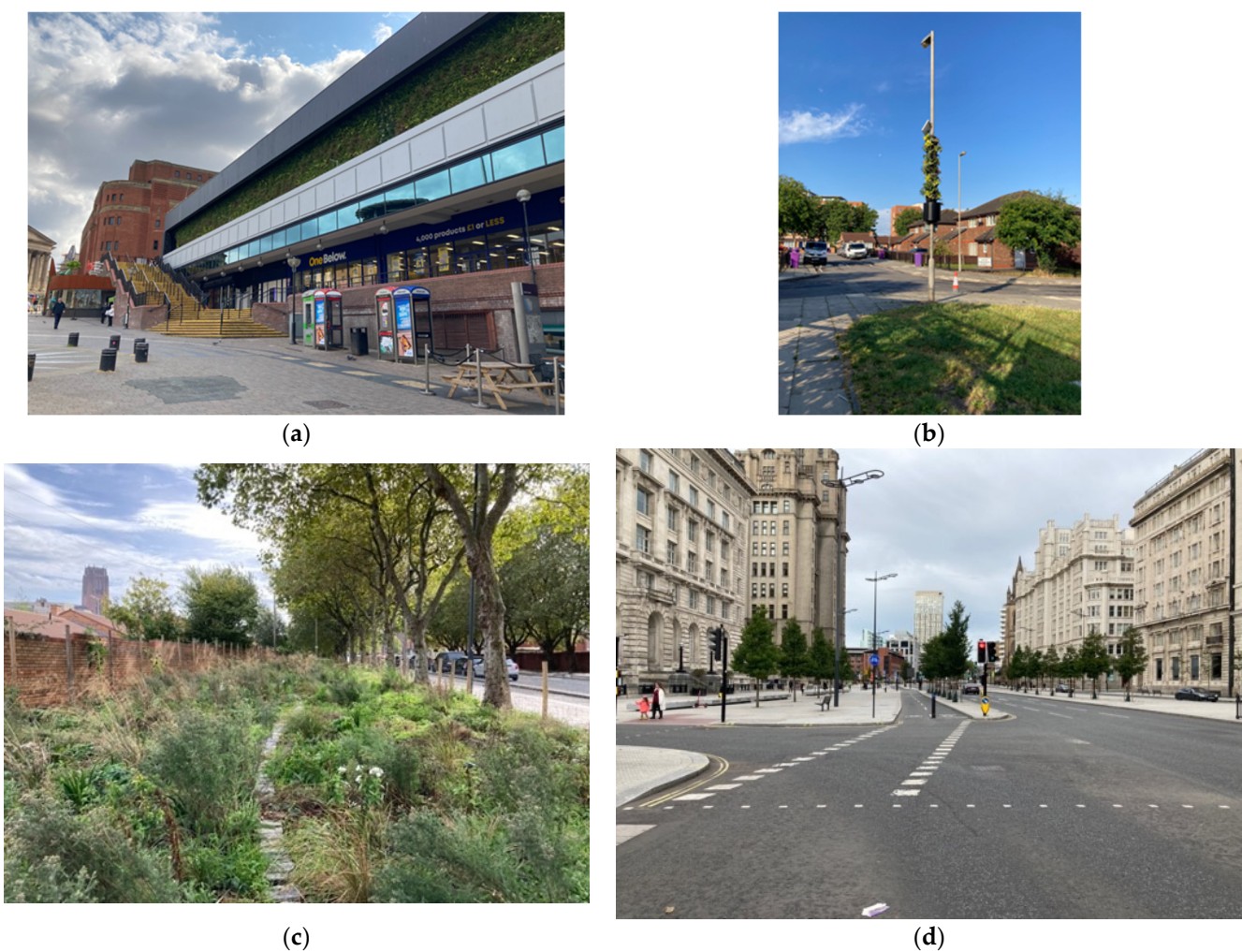

**Figure 1.** (**a**) NBS interventions in Liverpool—St John's Centre Green Wall, Liverpool City Centre. (**b**) NBS interventions in Liverpool—Pollinator lamp post installation, Baltic Triangle, Liverpool. (**c**) NBS interventions in Liverpool—Biodiverse pollinator planting, Park Lane, Liverpool (out of season). (**d**) NBS interventions in Liverpool—Street tree interventions to address traffic pollution and surface water flooding, The Strand, Liverpool.

Both policy and research on NBS underscore the need for participatory planning and co-design [26,27]. Over its lifespan, URBAN GreenUP worked extensively with local government, the environment and business sectors, technology start-ups/companies, engineers, landscape architects, and, to a lesser extent, local communities. The aim was to ensure that (a) the right NBS is located in the most appropriate place, (b) local stakeholders are aware of the NBS interventions and the associated socio-economic and ecological benefits, and (c) the range of benefits associated with each NBS intervention is grounded in robust evidence. This approach has been crucial in generating a detailed appreciation of what NBS interventions are needed, how they address local needs, and how they support local government policy mandates to address climate change, health and well-being, and economic development issues. The following reflects on the perceptions of local respondents to the changes afforded to the physical environment, their interaction with and valuing of urban nature, and the socio-economic and ecological benefits that investment in NBS can deliver. As a front-runner city within the URBAN GreenUP project, the city of Liverpool is being used as a testing ground for novel approaches to investment in NBS, and the acceptance and critiques of this program of work offer useful insights for cities adopting similar interventions. This study provides an analysis of public acceptance of NBS in terms of size, location, and type that can be used to shape future design and investment. It also

provides evidence of the role played by communication, engagement, and co-design of investment plans for cities, as well as a more nuanced appreciation of how stakeholders make links between alternative environmental and socio-economic factors. Evidence of this nature is valuable to cities in different locations and aids in the transferability of best practices (or the identification of poor practices) from which other locations can learn.

**Table 1.** URBAN GreenUP survey areas and associated NBS interventions.

| Survey Location | Description | URBAN GreenUP NBS Interventions |
| --- | --- | --- |
| **Business Improvement District** | City-center business area is characterized by a mixture of commercial, office, and retail space. Limited residential. Serviced via main roads. | St Johns/Parr Street/Chavasse Park Green walls, pollinator verges/green spaces, The Strand street tree investments, Wapping Dock Floating Ecological Island, mobile gardens/mobile forest bathing pods |
| **Baltic Triangle** | Area of mixed residential (mostly apartments but some Victorian terrace and social housing), light industrial, creative, and commercial spaces. Located next to the main road (Wapping, Chaloner Street, and Parliament Street) and River Mersey. Limited NBS on-site. | Street trees, pollinator lamp posts, pollinator verges/green space, forest school and church activities, sustainable drainage systems. |
| **Sefton Park** | Residential area of south Liverpool is characterized by a mix of apartments, semi-detached houses and converted Victorian townhouses. Some commercial/retail use. Sefton Park is the largest NBS in the area, a Green Flag awarded park, and one of Liverpool's most frequently patronized sites. | Floating Ecological Island, Street/shade trees, pollinator green spaces. |
| **Otterspool** | Residential area of south Liverpool is characterized by a mixture of 20th-century terraces and semi-detached houses. Otterspool Promenade and the greenspace system are the largest NBS in the area. Some light industrial use includes a neighborhood recycling center. Located proximate to A561 Aigburth Road and River Mersey. | Pollinator green spaces/verges, sustainable drainage systems, deculverting/bioretention flood work, wood/tree planting. |

The delivery of NBS will always be informed by historical debates and local politics, and this was certainly the case in Urban GreenUP. In Liverpool, green space is a marker of unequal investment and geographic inequalities. The wealthier southern parts of the city have more and higher quality GI and increased engagement with environmental issues, while communities in the north face multiple sources of social and economic deprivation and have less (and lower quality) GI [28]. Issues of spatial parity, environmental quality, access, and variability of amenities have been extensively debated and documented in the Liverpool Green Infrastructure Strategy [29] and the subsequent Liverpool Green and Open Space Review (LG&OSR). These indicated that NBSs in Liverpool are considered valuable and that greater investment in environmental management is needed to address climate change, biodiversity loss, a lack of sustainable transport options, and health and well-being inequalities [30]. One direct consequence of the LG&OSR has been the foregrounding of nature in the subsequent approach taken by Liverpool City Council to address environmental issues. It also facilitated Liverpool City Council's engagement with the EU's call for partners to join the Horizon 2020 NBS R&I projects and the city's inclusion in the URBAN GreenUP project.

A political willingness to engage with urban greening has thus been developed over several years. The relationships that Liverpool City Council has with regionally innovative environmental partners working on ecological issues are key to this. These institutions aided Liverpool City Council in engaging expertise to shape their environmental thinking by facilitating technical, academic, and knowledge exchange. However, despite the visible

relationships between the city, environmental organizations, developers, and the public, there remain critiques of Liverpool's environmental policies. These focus on the equitable provision of green space and the limited emphasis placed on tackling environmental quality issues compared to achieving economic development objectives. There is also a perceived lack of accountability associated with the City Council by some communities, which view all development as negatively impacting the city's natural environment [31]. The URBAN GreenUP project attempted to redress these issues by implementing a program of NBS interventions.

## 4. Materials and Methods

To evaluate the impact of NBS interventions, each frontrunner city developed indicators and a monitoring framework. In Liverpool, social indicators focused on understanding the knowledge, perceptions, and engagement with NBS among businesses, SMEs, environmental organizations and charities, residents, local communities of interest, i.e., church groups or friends of groups, and elected officials. Data were collected over an extended period (2019–2022) to ensure that a range of stakeholders who stand to benefit from the interventions were included.

The following sections analyze the evidence generated from two primary forms of data: interviews with communities of interest and the results of a postal survey (undertaken ex-ante and ex-post of the program of NBS interventions). Both approaches to data collection focused on communities of interest located proximate to URBAN GreenUP NBS interventions, as these respondents were considered to have a more detailed understanding of the local environmental context prior to and post-intervention (see Tables 1 and 2). However, the paper acknowledges that the sample size of the interviews and ex-ante/ex-post surveys is not statistically representative of the population of Liverpool or communities proximate to each intervention (see Table 3 for demographic profiles of proximate wards). The ex-ante survey aimed to establish a baseline position on perceptions of NBS in Liverpool, while the ex-post survey asked respondents to consider the URBAN GreenUP interventions and how these informed their perceptions of urban nature and its benefits. Data collection was influenced by COVID-19, which limited opportunities to engage with respondents face-to-face or on-site (see Section 4.3 for further details).

### 4.1. Interviews with Communities of Interest

A total of 22 semi-structured interviews were conducted with businesses, SMEs, social enterprises, non-governmental organizations' workers, members of 'Friends of' groups, and elected officials/councilors in Liverpool. Participants were selected due to their prior engagement with urban development and/or environmental issues prior to the commencement of URBAN GreenUP. Interviewees were located proximate to the intervention areas of Sefton Park, Otterspool, the Baltic Triangle, and the Central Business Improvement District (BID) or held a role of responsibility for their development, i.e., elected officials.

Practically, interviewees were provided information about the project, the nature of their engagement, and information regarding consent and anonymization prior to agreeing to engage. Interviews lasted between 30–90 min and were recorded and transcribed. Consent was obtained from all interviewees, enabling the project team to use their commentary in the public domain in an anonymized form. Each interview focused on the following:

(1)  Perceptions of the present provision of NBS/green space in Liverpool,
(2)  Perceived impacts URBAN GreenUP NBS interventions on the city's natural environment,
(3)  URBAN GreenUP governance structure and approach to urban greening, and
(4)  The perceived legacy of URBAN GreenUP.

The transcripts were analyzed thematically to illustrate where links were made between policy and governance, co-design, and engagement of local communities, the added socio-economic and/or ecological value of investing in NBS, and the potential for longer-

term benefits to Liverpool via the URBAN GreenUP interventions. Issues of design, engagement, focus of project interventions, long-term maintenance, and the delivery of benefits are key themes noted in the research literature focusing on NBS [8,32–34]. Direct commentary has been used from the fifteen participants (along with their organizations and areas of work—although twenty-two interviews were undertaken in total) shown in Table 2. Commentaries from the remaining seven interviewees corroborated the information presented but did not provide additional examples to extend the debates presented in the following sections. It is also acknowledged that the sample size of interviewees does not represent the wider body of professional organizations in Liverpool. However, the proximity of each organization and their knowledge provided interviewees with a greater understanding of the local urban and ecological context and the potential added value of URBAN GreenUP-funded NBS interventions. The project attempted to engage a larger number of organizations in the interview process. However, due to unforeseen logistical issues, organizations were not able to be involved in the data collection process.

**Table 2.** Interview profile and stakeholder type.

| Interviewee Profile | Stakeholder Type |
| --- | --- |
| Business-owner—Planning Consultancy | Business |
| Manager—Retail Organisation | Business |
| Owner—Environmental Consultancy Business | Business |
| Owner—Environmental Consultancy Business | Business |
| Business-owner—Hospitality | Business |
| Chair—Residents Organization | SME/Social Enterprise |
| Managing Director—CIC | SME/Social Enterprise |
| Manager—Natural Heritage NGO | NGO/NFP |
| Religious Leader | NGO/NFP |
| Head of Economic Non-Profit | NGO/NFP |
| CEO—Civil Society Organisation | NGO/NFP |
| Head—Parks Organisation | 'Friends of' group |
| Head—Parks Organisation | 'Friends of' group |
| Labour Councillor | Councillor |
| Green Party Councillor | Councillor |

*4.2. Ex-Ante and Ex-Post Postal Survey*

To ensure that a cross-section of local responses was generated, a resident's survey and expert/professional interviews were developed to assess local knowledge and use of NBS proximate to URBAN GreenUP interventions. The survey focused on respondent perceptions and relationships with nature in urban environments, their use of these spaces, and positive and negative assessments of NBS. It also reflected on how NBS could provide benefits to climate change mitigation, pubic/personal health and well-being, social inclusion, community engagement, the livability of the area, property values, crime, and local business opportunities. The questionnaire survey was constructed to provide respondents with opportunities to respond quantitatively via Likert/preference scales (a 5-point scale was used—strongly agree, mostly agree, neutral/neither agree nor disagree, mostly disagree, strongly disagree) and activity/issue lists, i.e., activities/uses of NBS, and qualitatively through open-ended questions. Both qualitative and quantitative questions were used to provide respondents with opportunities to detail their understanding of local NBS (Tables 4 and 5 provide indicative results of these types of questions).

**Table 3.** Ward Profiles of areas proximate to URBAN GreenUP NBS interventions.

| | Population | Male/Female | IMD (1 = Most Deprived to 30 = Least Deprived) | Unemployment Rate | Housing (Most Significant Tenure) |
|---|---|---|---|---|---|
| **Wards closet to Sefton Park and Otterspool NBS Investment Areas** | | | | | |
| **Church** | 13,722 | 48.02% (M)/51.98% (F) | 30 | 3.7% | 79.92% Owner Occupier |
| **Cressington** | 15,182 | 49.64% (M)/50.36% (F) | 24 | 5.2% | 69.2% Owner Occupier |
| **Greenbank** | 15,731 | 47.73% (M)/52.27% (F) | 22 | 6.1% | 48.73% Private Rented |
| **Mossley Hill** | 13,463 | 49.91% (M)/50.09% (F) | 29 | 3.5% | 73.9% Owner Occupied |
| **Princes Park** | 20,529 | 47.67% (M)/52.33% (F) | 8 | 12.8% | 49.31% Registered Social Housing |
| **St. Michaels** | 12,724 | 47.33% (M)/52.67% (F) | 20 | 7.0 | 47.46% Owner Occupier |
| **Wards closest to BID and Baltic Triangle NBS investment area** | | | | | |
| **Central** | 33,468 | 46.92% (M)/53.08% (F) | 26 | 2.3% | 78.67% Private Rented |
| **Princes Park** | 20,529 | 47.67% (M)/52.33% (F) | 8 | 12.8% | 49.31% Registered Social Housing |
| **Riverside** | 23,498 | 47.77% (M)/52.23% (F) | 15 | 7.3% | 54.89% Private Rented |

**Table 4.** Survey perceptions of NBS quality, quantity, and accessibility located proximate to URBAN GreenUP interventions (ex-ante and ex-post results). Boxes in red denote an overall negative response and those in green an overall positive response.

| | Sefton Park (Ex-Anti Intervention) | Sefton Park (Ex-Post Intervention) | Change in Response (Positive/Negative) | Otterspool (Ex-Anti Intervention) | Otterspool (Ex-Post Intervention) | Change in Response (Positive/Negative) |
|---|---|---|---|---|---|---|
| **How would you rate your neighbourhood NBS in terms of quantity** | 93.7% Good/Very Good | 92.1% Good/Very Good | Negative | 100% Good/Very Good | 97.6% good/very good | Negative |
| **How would you rate your neighbourhood NBS in terms of quality** | 84.7% Good/Very Good | 82.5% Good/Very Good | Negative | 90.5% Good/Very Good | 89.3% Good/Very Good | Negative |
| **In your neighbourhood, how would you rate the NBS in terms of accessibility.** | 90.7% Good/Very Good | 89.6% Good/Very Good | Negative | 85.7% Very Good/Good | 92.8% Good/Very Good | Positive |
| **Thinking about the city of Liverpool as a whole, how would you rate its NBS in terms of quantity.** | 69.9% Good/Very Good | 64.7% Good/Very Good | Negative | 76.2% Good Very/Good | 65.9% Very Good/Good | Negative |
| **Thinking about the city of Liverpool as a whole, how would you rate its NBS in terms of quality.** | 65.6% Good/Very Good | 65.6% Good/Very Good | Negative | 54.7% Very Good/Good | 75.3% Very Good/Good | Positive |
| **Thinking about the city of Liverpool as a whole, how would you rate its NBS in terms of accessibility.** | 61.3% Good/Very Good | 59.2% Good/Very Good | Negative | 65.9% Very Good/Good | 70% Very Good/Good | Positive |

**Table 5.** Survey respondent knowledge of URBAN GreenUP NBS interventions (drawn from ex-post survey only). Boxes in red denote an overall negative response and those in green an overall positive response.

| | Sefton Park (Post) | Reaction (Positive/Negative/Neutral) | Otterspool (Post) | Reaction (Positive/Negative/Neutral) |
|---|---|---|---|---|
| Have you seen the green wall at St. John's Shopping Centre? | 65.1% No | Negative | 82.4% No | Negative |
| Were you aware that it was an URBAN GreenUP intervention? | 84.9% No | Negative | 96.3% No | Negative |
| Have you seen the green wall on Parr's Street? | 81% No | Negative | 90.4% No | Negative |
| Were you aware that it was an URBAN GreenUP intervention? | 91.8% No | Negative | 97.6% No | Negative |
| Have you seen the floating island in Wapping Dock? | 79.1% No | Negative | 91.8% No | Negative |
| Were you aware that it was an URBAN GreenUP intervention? | 81.9% No | Negative | 97.6% No | Negative |
| Have you seen the floating island in Sefton Park? | 69.8% Yes | Positive | 56.5% Yes | Positive |
| Were you aware that it was an URBAN GreenUP intervention? | 74.4% No | Negative | 81.5% No | Negative |
| Have you seen the bio-retention pond in Otterspool? | 53.5% yes | Positive | 71.8% yes | Positive |
| Were you aware that it was an URBAN GreenUP intervention? | 90.7% No | Negative | 89.3% No | Negative |

Surveys were distributed to homes within a 300 m radius of the two research sites: Sefton Park and Otterspool, located in south Liverpool and hosting URBAN GreenUP NBS Interventions. Two surveys were conducted—the first in 2019 prior to NBS interventions to establish a baseline of local perceptions of NBS (hereafter the ex-ante survey) and a second in 2021 following the completion of NBS interventions (hereafter the ex-post survey). The postal survey was delivered by hand and collected by the research team because (a) postal surveys without interaction with the research team generate lower returns and (b) they provided the research team with an additional level of certainty regarding how many people/homes had been engaged, how many had responded, and whether additional visits to collect/remind residents to complete the survey were needed. Respondents were also provided with the opportunity to return completed surveys via postal mail in a pre-paid envelope. The use of a postal survey was considered to add greater validity to the data collection process, as the research team could guarantee (within some tolerances) that the survey would be completed by people living proximate to existing and new NBS interventions. Respondents were able to complete the survey by hand for collection by the

project team and return it by pre-paid envelope. They were also able to complete it online via a designated survey weblink. Details of the online survey (including a weblink) were included in the information provided in the postal survey.

A total of 75 survey responses were received from residents in the ex-ante survey (N: 35 Sefton Park and N:42 Otterspool), and 173 responses (N:86 Sefton Park and N:87 Otterspool) were returned following the ex-post survey. The combination of data collected for the survey provided scope to analyze whether links between preferences, uses, appreciation and/or understandings of NBS in Sefton Park and Otterspool could be made.

### 4.3. COVID-19

URBAN GreenUP ran from 2017–2023, and the main period of data collection occurred during the 2019–2021 COVID-19 pandemic. Consequently, the approach taken to data collection was modified to adhere to institutional and UK government stay-at-home and social distancing regulations. The outcome was that face-to-face surveying work was deemed inappropriate, and data collection was predominately conducted online, i.e., interviews on Zoom or via postal survey, where interaction with the research team was minimal. The authors acknowledge the implications of such a pivot from face-to-face evidence collection in terms of potentially limiting the number and range of potential participants engaged. However, the alternative methodological framework developed was considered robust, providing an effective approach through which residents and communities of interest could be engaged.

### 5. Results

The outcome of interviews with local communities of interest and the postal survey with residents illustrate a range of considerations regarding the value of both NBS generally in Liverpool and URBAN GreenUP NBS interventions specifically. Results have been categorized and presented in three distinct areas: Elemental and functional influences, Project and locational influences, and Thematic influences of NBS success to highlight the variation in participant responses related to the breadth of views that need to be made when implementing NBS.

The presentation of the results is not split between interviewee and respondent responses but is presented thematically across the three areas noted above. This is a deliberate choice, as it is considered to show greater complementarity between the different voices engaged with NBS. Sections 5.1–5.3 combine evidence from the interviews and survey results, as both address the thematic framings of each sub-section. This provides the discussion with a more nuanced approach to comparing issues deemed significant to professionals and residents. Direct quotes or specific points are attributed to specific interviewees or noted as being drawn from the resident survey.

The data generated from the interviews are also presented as a set of critical commentary reflecting participant considerations of the additional value that NBS and the URBAN GreenUP interventions can provide in Liverpool. The survey responses were analyzed with descriptive statistics to explore trends in the data. In Tables 4 and 5, for example, the data provided is drawn from Likert Scale analysis and depicts the most frequently used responses to a series of questions focused on perceptions of NBS quality, quantity, distribution, visibility, accessibility, and use. It is also acknowledged that there is a cross-over between both the results derived from the interviewee and survey responses, as investment in NBS does not fall solely into the purview of professionals or residents and, as such, should not be compartmentalized when issues of politics, sociocultural, economic, and environmental issues intersect.

### 5.1. Elemental and Functional Influences: Location, Scale, Functionality, Interactivity

Significant emphasis was placed on the location, scale, and quality of NBS in Liverpool by both respondents to the postal survey and interviewees. From an interviewee's perspective, the city has a host of high-quality NBS in the form of its Victorian parks and

waterfront area; however, it was noted that the distribution in terms of north-south, areas of low income/affluence, and proportion of these spaces was inequitable. This was reported by an elected official, who stated that:

*"I think we have some wonderful parks in the city, particularly in the south. They are not proportionate. I think we need more green corridors, whether they are pedestrian or bike lanes, that people can use."*

The view that NBSs are not distributed equally was also commented on by a business respondent when considering access to nature in urban areas: *"I have absolutely no idea what the people who live in the city are doing for their walks and to get out of the house because they have not got anything that is remotely connected at the moment."* However, an alternative and more positive elected official commented that URBAN GreenUP was facilitating a conversation about the breadth of options available to Liverpool City Council and partners to invest in NBS. They stated that:

*"Scientifically, I do not know how much of an impact [URBAN GreenUP NBS interventions are] going to have because, obviously, they are fairly small [but] something like that [floating island] I think is useful as a talking point. Then we can start the conversation about biodiversity. My attitude is that you should try and work out what the problem is and what you want to do about it, then how you do it, and then find the money. This [URBAN GreenUP] really started with the money, which is not a very good way to start. It is very difficult for a politician or a senior officer to resist a project that comes with money attached."*

Therefore, although issues of location and scale were reported, URBAN GreenUP could act as a catalyst for the city to rethink how it invests in urban nature by reflecting on what type of intervention would work in specific locations.

Considering NBS from such a perspective would enable the City Council (and partners) to address the pervasive view that there is insufficient NBS across Liverpool. Table 4 highlights this issue: Postal survey respondents had largely positive views of NBS (85%+ positive responses for local NBS), but views of the city's environmental resources were less positive. Responses to the postal survey vary in their assessment of the quality, quantity, and accessibility of NBS locally and at a city scale. Local/neighborhood NBS are perceived more positively than Liverpool's. This aligns with expert commentary in the city and academic literature and highlights an ongoing issue of environmental governance within Liverpool related to parity of resource allocation.

By contrast, the views expressed in the ex-post survey are also conditioned by responses to COVID-19 and city-wide lockdown protocols. As such, perceptions of accessibility, quality, and quantity may have altered from 2020 onward if respondents had (a) limited access or (b) spent more time in green space and thus were more critically aware of quality/quantity issues. Moreover, it could be argued that respondents in Sefton Park are more critical of the totality of Liverpool's NBS resource base due to their proximity to one of the city's two Green Flag accredited parks (Sefton Park and Stanley Park). Residents in Otterspool may not have placed such an emphasis on local sites in their commentary. Therefore, they may have been more accepting of the variation in location, size, and function due to the greater variation in NBS quality compared to Sefton Park residents. Otterspool residents report improvements in accessibility locally (an increase from 85.7% to 92.8%) and across the city (an increase from 54.7% to 75.3%) in the ex-post survey, supporting this view.

The variation in commentary from local expert interviews and residents supports existing discussions concerning the distribution of NBS in Liverpool. This highlights the need for continued engagement with issues of how projects identify sites for development (as well as financing for maintenance) across the city. It also suggests that although URBAN GreenUP has value for enhancing ecological resources, the interventions may be inadequate, especially for addressing the existing disparity caused by long-term development objectives across the city.

*5.2. Project and locational influences: Communication, Visibility*

> *"Firstly, I think individuals, businesses, and developers have not gotten a clue. They need something like this [URBAN GreenUP] to give them ideas . . . they know about buildings, and they know about commerciality, but they have no clue about how they can do it in a very considered way. There is a lack of creativity in terms of what we have here, which is a bunch of facilities that could have green all over the place in terms of their roofing. There is a real lack of imagination in terms of how green space can actually be created. We are not being individually or collectively clever enough."*

The lack of visibility of the URBAN GreenUP interventions was perceived as undermining their success by interviewees in business, the environment, the SME sector, and local government. The quote above from a local NGO highlights two significant issues with NBS interventions: (a) local awareness of project work and (b) the type of projects delivered. The NGO officer critiqued the lack of innovation by local businesses, community, and environmental organizations in terms of experimenting with NBS beyond planting street trees, street greening, or creating pock parks. This raises questions regarding whose responsibility it is to facilitate innovation and/or investment in NBS and who has the authority to deliver change in urban areas where multiple landowners are present. Even where substantial interventions were delivered, i.e., the 65 m long "living green wall" comprising over 14,000 evergreen plants on the St. Johns Shopping Centre, there was a view from a local Small–Medium Sized Enterprise (SME) that *"[local people] will not see the fringe [the green wall] around St. John's Shopping Centre in any way making up for some of the stuff that is going on."* Although a further SME noted that they considered URBAN GreenUP to have support within the city, this was skewed towards the environment sector rather than from a broad cross-section of stakeholders:

> *"I think URBAN GreenUP as a concept has support, but only among a small group of people in the know, predominantly eco-warriors and university types, and people with responsibility within the council for eco-friendly decisions."*

One issue repeatedly noted by interviewees extending this view is the city's role as the facilitator of URBAN GreenUP and the language used to raise awareness of the project. The term "NBS" is considered overtly technical and thus lacks resonance with local people. As noted by a representative of a 'Friends of' Group, such language makes NBS less tactile, as it is not as common compared to parks, trees, or nature (although each of these terms is also complex). Moreover, within the NGO and environmental sectors, two respondents commented:

> *"I would not naturally have put that phrase [NBS] with URBAN GreenUP unless you are defining the problem as there is not enough green space and we would like it to be more . . . I would have gone more for something like nature-based enhancements. If it is more about how we actually want to enhance green corridors and make it a nicer place to live, I would not necessarily think 'solution' was the phraseology for that."* (NGO)

> *"I have a personal reaction to the word 'solution', and it is not a good one because a solution emanates from a problem, and I just do not like problem–solving processes; they are too simplistic. You take a problem and you solve it; you [then] usually create ten more problems, often they are somebody else's, and it flows on from there, and sometimes they come back to bite you. It is an endless task. It is also symptomatic of simplifying quite complex systems."* (Environmental consultant)

Both highlight a potential limitation of NBS terminology if it does not readily translate to all communities. Moreover, the comments illustrate how, even within the environmental sector, interpretations vary regarding the meaning of words. In the case of NBS, the use of "solutions" is viewed as problematic if or when interventions are not linked directly to context-specific issues. In Liverpool, the city's Green Infrastructure Strategy [26] and the URBAN GreenUP project proposals identified a range of core climatic, socio-economic, and health issues that NBS could address. However, although this information

was integrated into the framing and project documentation, development, and delivery of URBAN GreenUP, it may not have been effectively communicated to all communities in the city—especially residents (see Table 5, which presents resident awareness of URBAN GreenUP interventions).

Consequently, there were concerns among a significant number of participants regarding the communication of the added value of NBS interventions associated with URBAN GreenUP. This is also visible in the responses from the ex-ante and ex-post postal surveys, where awareness of both the project and its specific interventions was limited. Table 5 outlines the awareness of Sefton Park and Otterspool respondents, noting that the floating ecological island in Sefton Park and the bioretention work in Otterspool were the only known interventions. This suggests that the interventions delivered were (a) physically too small to be visible to residents, (b) not well-publicized, or (c) in places that people do not use. Point (c) is countered by a significant number of respondents in the ex-post postal survey who reported using Sefton Park and Otterspool frequently as their local NBS/greenspace compared to other city center or waterfront locations and said that the URBAN GreenUP interventions in these locations may have been increasingly visible compared to those in the Baltic Triangle or BID area. This was noted by a Friends of Group located close to Sefton Park, who stated that:

> "The impact was immediate in terms of literally every person passing us . . . stopping to ask what it was about. We did not receive one comment about what a waste of money 'in these times' [they were], which I was quite surprised at. I did assume you would be receiving, 'Oh, well, why are you spending money on this when, you know, we have poverty and COVID and everything else?' [There has been] lots and lots of genuine interest, which was surprising in terms of impact."

It could be argued that the communication of each intervention could have been improved to facilitate knowledge of both the URBAN GreenUP project and the NBS innovations within it. The commentary from local environmental organization interviewees supports this, suggesting that consideration of terminology and engagement with communities of interest are key to raising awareness. Therefore, projects need to be visible to users to facilitate engagement. Where this was possible (e.g., the water management work in Otterspool), users saw work on-site and a direct impact (less flooding) of the intervention. Thus, the experiential nature of specific projects in highly visible locations was deemed necessary in assessing the value of NBS interventions as "solutions".

### 5.3. Thematic Influences of NBS Success: Ecological and Socio-Ecological Factors

The postal survey results highlight a positive response to assessments of the environmental quality associated with the URBAN GreenUP interventions. Although knowledge of each intervention was less well defined in responses (see Table 5) when residents were asked to discuss the links between NBS and climate change, biodiversity enhancement, and quality of life, respondents in both the Sefton Park and Otterspool surveys provided a positive analysis of the resource base in Liverpool. NBS interventions were considered to have a less direct impact on addressing urban heat island effects and improving air quality. Although respondents understood they could make a difference in these problems, NBS interventions were perceived as more effective for other environmental challenges. NBSs were perceived as having the least impact on economic factors, with resident survey respondents skeptical of linking nature and local business revenue. The lack of positive responses for economic activity was in line with the general commentary from residents in the postal surveys, as they focused more directly on issues of quality of life and place than economic returns. A reading of Table 6 suggests that NBS are considered to positively contribute to the quality and functionality of ecological resources in Liverpool—noted as strongly agreeing or somewhat agreeing in responses to postal survey questions. However, there was a minor proportion of responses considered neutral or negative overall, i.e., instances of respondents strongly disagreeing with links between NBS and improved environmental quality. Drawing on the ex-post postal survey, Table 6 notes the prominence

of improvements in quality of life and biodiversity (and, to a slightly lesser extent, climate change) as positively associated with NBS interventions.

**Table 6.** Postal survey responses to the impact of URBAN GreenUP NBS interventions. Green boxes denote positive overall associations between NBS interventions and specific benefits, whilst yellow boxes denote a partial relationship between identified in respondent commentary.

| | Sefton Park | | | | | | Otterspool | | | | | |
| | Impact of URBAN GreenUP Intervention (Respondent Commentary—Green = Positive/ Yellow = Neutral/Red = Negative) | | | | | | Impact of URBAN GreenUP Intervention (Respondent Commentary—Green = Positive/ Yellow = Neutral/Red = Negative) | | | | | |
| | Climate Change | Air Pollution | Urban Heat Island | Local Business Revenue | Quality of Life | Biodiversity | Climate Change | Air Pollution | Urban Heat Island | Local Business Revenue | Quality of Life | Biodiversity |
|---|---|---|---|---|---|---|---|---|---|---|---|---|
| **St Johns Shopping centre green wall** | Yellow | Green | Green | Yellow | Green | Green | Green | Green | Yellow | Yellow | Green | Green |
| **Parr Street green wall** | Green | Green | Green | Yellow | Green | Green | Green | Green | Green | Yellow | Green | Green |
| **Floating Island Wapping Dock** | Yellow | Yellow | Yellow | Yellow | Green | Green | Yellow | Yellow | Yellow | Yellow | Green | Green |
| **Floating Island Sefton Park** | Green | Yellow | Yellow | Yellow | Green | Green | Yellow | Yellow | Yellow | Yellow | Green | Yellow |
| **Bioretention Pond Otterspool** | Yellow | Green | Green | Yellow | Green | Green | Green | Green | Green | Yellow | Green | Green |

Absent from Table 6 is a commentary on the economic opportunities associated with NBS. These data were generated from interview respondents in the BID and Baltic Triangle rather than the postal survey. Interviewee respondents reported that businesses responded positively to NBS as a facilitator of economic development opportunities. The commentary noted that greener, interactive, and ecologically diverse environments attracted both businesses and supported additional footfall and returns on investment, justifying relocation costs to "greener areas." Respondents also noted links between higher rental and sales values in locations with more NBS. They supported the view that employee productivity would increase if and where neighborhoods had a higher proportion of NBS. All of these compare favorably to the research literature examining the links between NBS/GI and economic value [32,33,35]. To support these statements, an SME and a local business representative noted:

*"A city that is greener, with a lot more nature in it . . . is a much more attractive place to work and to attract businesses. It is a healthier place to work. Therefore, during breaks, you can get out, and very quickly, you are under a tree. You are looking at flowers; you are sitting on a bench where there is some grass around; there is a water feature, or whatever it is. That, of course, impacts the people who come into the city to work, the visitors to the city, and the people like us who live here; it impacts significantly on our well-being and our health."* (SME)

*"Inward investors are more likely to choose greener cities for their businesses and employees."* (Local business)

However, the need for caution was also noted in terms of over-extrapolating the impact of urban greening/NBS on city-wide economic development opportunities. A local business reported that:

*" . . . businesses use many criteria when deciding where to locate, and many different factors play a part, with green spaces not necessarily high on the list. Transport, parking, and cost would be far more important."*

Interviews with business respondents also provided a more detailed analysis of the perceived links between NBS and the economic benefits of investment. This included a

more nuanced appreciation of NBS, as it was more difficult for respondents to substantiate claims of direct economic benefits associated with NBS interventions. Alternatively, business respondents outlined how they considered NBS to aid this process, arguing that, for example, 80% of respondents in the BID and 50% of respondents in the Baltic Triangle reported that investment in NBS would lead to an uplift in property prices. Increased opportunities for employment (based on NBS providing a more attractive work and investment environment) were also noted: 50% agreed/strongly agreed in the BID and 75% agreed/strongly agreed in the Baltic Triangle. There was also consensus that locations with a higher proportion of high-quality, diverse, and interactive environments, including NBS, had the following effects: (a) they are attractive to businesses (83% agree/strongly agree in the BID and 63% agree/strongly agree in the Baltic Triangle), (b) they promoted investment and relocation these areas (66% agree/strongly agree in the BID), (c) they support increased footfall and time spent leading to potential increases in revenue (83% agree/strongly agree in the BID and 50% agree/strongly agree in the Baltic Triangle), and (d) they are greener and more interactive places can facilitate increased productivity (83% agree/strongly agree in the BID and 63% agree/strongly agree in the Baltic Triangle) and employee well-being. However, while the majority of respondents identified positives associated with NBS, there remained concerns that parking, public transport, and the cost of rent were more significant influences on economic development opportunities than investment in NBS. Additionally, businesses in the Baltic Triangle stated that NBS was not a core factor promoting their relocation to the area (63% disagree/strongly disagree).

Consequently, no singular view supports investment in NBS as a facilitator of socio-economic or ecological improvements. What is visible is the complexity of the benefits and/or functions respondents (residents and interviewees) associate with NBS, what they highlight as positive influences, and where they view change as having a less significant impact. Therefore, defining success in delivery is difficult, as no singular function or intervention was deemed to deliver improvements in socio-economic or ecological benefits in all cases. This suggests that when planning NBS, local context is critical to the choices being made, which requires awareness of the perceptions of businesses, the environment, community-oriented organizations, and residents to ensure investment responds to local circumstances. Each of these groups defines success differently, and as such, the city needs to be aware of how scale, focus/function, and visibility influence perceptions of successful interventions. Although there was a broad consensus among the respondents engaged with the project that NBS can enhance the quality, quantity, and functionality of a local area, the use of NBS should not be seen as a panacea for all urban problems—especially economic issues.

## 6. Discussion

Investment in NBS has been framed as a "go-to" approach to addressing cities' complex socio-economic and ecological problems via co-produced nature-centric plans [36]. NBSs offer a breadth of investment options that can be adopted to address issues including ecological decline, climate change, unsustainable urban forms, and socio-economic deprivation [21,37]. Investment in NBS by the EU, funded via the Horizon 2020 NBS R&I program, accelerated implementation and facilitated experimentation with NBS, which may not have occurred or been delivered over an extended timeframe via ES or GI planning. This has been particularly visible in the support for NBS from local government, environmental organizations, and SMEs, who have been at the forefront of these delivery programs. Consequently, we can identify a groundswell of engagement with NBS as potential "solutions" to urban problems that are flexible enough to adapt to various contexts.

A significant proportion of the literature supporting this position frames NBS as an evolution of other nature/green space planning terms, but one that explicitly looks to deliver change. This is becoming clearer in the literature debating scoping, design, implementation, and monitoring of NBS interventions in various European, Chinese, and African cities. A further commonality across these debates is the framing of NBS

as a facilitator of more effective human/environmental discussions—one where nature offers direct solutions to human (and ecological) problems [19,34,38]. Within the URBAN GreenUP project, links between the ecological benefits and the associated socio-economic opportunities afforded by NBS interventions have been central, e.g., enhancing recreation, access to nature, and economic development [39]. Against this backdrop, each of the five-year Horizon 2020 R&I programs tested a range of delivery options to examine how NBS can create more sustainable and functional urban environments. This framing can be mapped to the approach taken in Liverpool.

Discussions of the added benefit of NBS require an understanding of (a) NBS form and function, (b) the benefits delivered to people and biodiversity, (c) the role of NBS as inclusive long-term solutions, and (d) the contexts in which they are implemented, and such benefits need to be communicated effectively to all communities of interest [6]. In practice, this requires dynamic, iterative planning processes to ensure effective delivery in the nuanced circumstances of different locations. Liverpool, for example, focused extensively on climate change adaptation, access to and improved quality of landscape functionality, and economic development opportunities—A, B, and C above. However, the approach to communication and context-driven delivery was queried by the respondents to the survey work undertaken for URBAN GreenUP.

However, analysis of NBS investments in Liverpool enables advocates to identify whether these findings can be mapped onto the growing series of frameworks developed for NBS to provide signposts for delivery. The research of Frantzeskaki [40], for example, reported a further seven areas that should be considered when developing NBS: NBS should be attractive, help create new green commons, promote trust between the public and city officials, support collaborative governance, facilitate inclusive and holistic policy formation, and be scalable and transferable between locations. All of these are common to each Horizon 2020-funded NBS project and are identified as key factors influencing the framing of NBS in the literature. The evidence discussed above and the wider reporting on the Horizon 2020 NBS projects provide opportunities to reflect on what best practice for NBS looks like, even if the NBS interventions delivered in Liverpool do not fully align with these goals [19,39–43].

*6.1. Locating NBS in Local Development Structures*

In the context of Liverpool and the delivery of NBS via the URBAN GreenUP project, a continued reflection on issues of appropriateness in terms of the location of project interventions, collaboration and trust between stakeholders, and the creation of additional ecological resources in urban areas were all considered critical. Critiques can be made of the URBAN GreenUP portfolio of interventions, including whether they were the most effective way of delivering change. While the analysis shown above suggests not, the commentary of professional and residential stakeholders supports the links between environmental and socio-economic improvements associated with the project. This was significant when issues of access and use of NBS and perceptions of quality of life were discussed by respondents. However, NBS advocates must remain cognizant of the local context to ensure effective delivery. Otherwise, concerns about legitimacy could be raised, e.g.,

> *"I get the sense that it is more greenwashing than actually addressing urban design problems, living problems, and urban-ecological problems of the human species."* (Environmental Consultant)

> *"While I am generally supportive and positive about the URBAN GreenUP project, my critique would be that what something like URBAN GreenUP represents is this idea that you have to raise lots of money, spend loads of time consulting and planning, and then spend loads of money implementing a complicated big project to encourage nature. What you need to do is just stop wasting your time fighting nature the whole time and allow nature to flourish. Nature can do a pretty good job on its own."* (Elected official)

URBAN GreenUP may have succeeded in extending the foundations of the city's Green Infrastructure Strategy and the LG&OSR [25–27] by building additional momentum for interventions in nature-led planning that can subsequently be used to shape policy. In such a scenario, NBS can be used to explore new delivery pathways in terms of innovative design, the choice of projects and the location they are placed in, and the role of experimentation in terms of the benefits associated with each NBS, which provides scope to rethink environmental planning across the city [44–46]. If this can be effectively achieved, Liverpool City Council could inspire further interaction with nature while promoting the efficient use of existing space within the city for development [47]. The evidence generated by URBAN GreenUP could be beneficial if leveraged to support an increased allocation of local government capacity and financing to facilitate longer-term changes in the governance of the city's landscape.

### 6.2. Elements, Functions, Benefits, and Beneficiaries

To ensure that NBS interventions deliver their intended objectives, there is a need to consider what elements are designed, what functions, i.e., specific ecosystem services, they deliver, what socio-economic and/or ecological benefits they provide, and who benefits from investment in nature. Raising awareness of URBAN GreenUP interventions and other greening projects is important as a starting point. Despite the visibility of some NBS interventions, i.e., the Sefton Park floating island, there remains a lack of connection between policy/local government campaigning for NBS and public acknowledgment of the added benefit of the interventions. Project partners in Liverpool and elsewhere could benefit from more attention being placed on the needs and aspirations of the diverse communities that stand to benefit from NBS interventions. The promise that NBS can deliver more democratic and sustainable outcomes is central to arguments for their use [26]. The "NBS with and for people" approach can help deliver on such promises, enhancing equitable distribution of benefits, minimizing disbenefits, and underpinning successful delivery [48]. It could be argued that URBAN GreenUP did not fully align the needs of people, place, and nature, leading to critiques of the approach.

Working with communities to respond to locally contextual issues also enables NBS supporters to think more creatively about a given intervention's type, size, function, and benefits. Moreover, it provides scope to consider how variations in aesthetics and ecological diversity can be more effectively aligned with issues of accessibility to promote use [40,49]. URBAN GreenUP highlights that this is not a simple process and requires ongoing collaboration between multiple sectors. If such relationships can be curated, all parties can better engage in knowledge exchange activities and arrive at more appropriate nature-focused solutions [26].

### 6.3. Aligning Political and Local Needs/Priorities

Evidence from Liverpool suggests that in addition to considerations of location and function, advocates of NBS also need to align interventions with local political and planning objectives. The diversity of commentary from local elected officials noted above is in keeping with historical debates about the environment in the city [25,28]. This highlights the critical role of consensus (or lack thereof) in shaping investment plans and political priorities. Where the breadth of benefits associated with NBS, e.g., enhancing ecological functionality alongside economic development, can be integrated into policy mandates, the options open to the City Council and its partners expand. Achieving this requires the city to examine the true cost of 'business as usual' development and reflect on the benefits of shifting policy attention from predominately built infrastructure to a more NBS-centric approach. This is discussed extensively in the literature and demonstrates a need to debate the variability and complexity of alternative governance models to ensure that expertise, innovation, and local knowledge are integrated into decision-making [2,10,11,16,19,20,34,50]. Where each can be aligned effectively, we can identify investment options for NBS that are locally appropriate, as well as new funding pathways and alternative management practices that make the best

use of local government, private and environmental institutions and organizations, and community advocates to deliver investment in NBS [33,34,39,40].

Moreover, by examining NBS interventions as a continuum of options linking local, city, and regional perspectives, cities can better identify the most appropriate approaches for investment [51]. Although the URBAN GreenUP project was focused predominately on the local scale, there are options to explore the delivery of comparable innovations at a larger scale due to the scalability of investment in street trees, sustainable drainage, and pollinator networks. This can even be extended to a national scale if the priorities of such policies, for example, the National Planning Policy Framework in England [52] or EU directives on NBS [50], can be aligned with local delivery. This provides greater scope to promote synergies between policy mandates, identify problems associated with resource change and/or technological redundancy, develop adaptive plans to address maintenance and redevelopment issues and assess the immediate and long-term benefits associated with urban change [53].

### 6.4. Communication and Visibility

Analysis of the NBS interventions in Liverpool also illustrates the critical role played by communication and visibility in developing successful outcomes for investment. Across all commentary, the lack of communication of NBS projects delivered as part of URBAN GreenUP was noted as problematic for communities of interest. Therefore, effective communication of the proposed NBS elements, their functions and benefits, and the beneficiaries associated with interventions must be clearly communicated to all [6]. This is a critical point, as although NBS continues to deliver ecological services (e.g., flooding mitigation, providing habitat, intercepting pollution) even when people are unaware of these functions, the lack of awareness of the benefits can undermine public acceptance of investment in urban nature. Successful communication can support a greater understanding of the functions associated with NBS where they are not highly visible, which in turn can lead to public support for future interventions. It is critical to develop a successful communication strategy highlighting the location, function, and NBS to increase their visibility.

### 6.5. The Future Role of Stakeholders as Advocates for NBS Investment in Liverpool

The limited role afforded to environmental experts, local practitioners, businesses, and communities during the delivery of the URBAN GreenUP was reported as limiting the perceived value of the project's investment program. Future work should thus consider engaging with local expertise from experts and residents to help shape considerations of what type and size of NBS and in which locations investments should be delivered. Specifically, environmental stakeholders have expertise in landscape design, funding, and maintenance, which could be integrated into project work. Moreover, business leaders are acutely aware of the links between urban/landscape quality and economic prosperity and could be useful commentators on the appropriateness of future NBS interventions. Therefore, stakeholders engaged with urban development and management may be critical allies for the city council in Liverpool if or when they design, plan, and implement additional NBS. In addition, consultation with residents will be a key activity needed to ensure that the type of NBS, scale of intervention, and location of investment are considered appropriate to the local context. The city of Liverpool is aware of the need to engage more effectively with its stakeholders, as reported in the Liverpool Green and Open Space Review [30]. Still, it has not yet established a clear structure to mee this challenge. It may therefore be appropriate for local stakeholders to be afforded a more prominent role in the design, implementation, and management of NBS across the city of Liverpool. However, to date, the city council does not have a governance framework in place to ensure that such a multi-partner approach to investment is implemented. A key outcome of URBAN GreenUP is the need to rethink the role of a diverse set of stakeholders in the city's environmental planning and management.

*6.6. Reflection on Limitations*

The public and stakeholders perceived NBS as valuable and useful. Still, the methods were not designed to access a deep understanding of public knowledge, values, and perceptions of the benefits and drawbacks of NBS. If monitoring had been better resourced, surveys would have been combined with more deliberative methods, as surveys do not provide scope for respondents to elaborate. If the project were completed again, it would also aim to generate a more representative sample of the local population constructed by age, housing type, employment, and income. The survey would also add additional questions to generate baseline knowledge of participant awareness of benefits and disservices to compare responses to the URBAN GreenUP NBS. Due to the limitations placed upon the project by time, staffing, and COVID-19, this was not feasible.

The results need to be contextualized in light of the impact of COVID-19, but the direction of influence is potentially positive and negative. The pandemic saw an increase in the use, considerations of accessibility, and perceptions of value placed on urban nature [54,55]. This could have been reflected in our results, as the majority of respondents in the postal surveys and interviews noted a greater appreciation of the sociocultural, ecological, and economic value of NBS post-lockdown. Although this may not translate directly into new or revised policy supporting investment in NBS/greenspace, it provides valuable evidence of the relationships between people and their environments. Such an outcome compares favorably with the results of analysis by Public Health England [56] and Natural England [37,38] in their analysis of greenspace use, suggesting that further funding for nature is key to healthy and successful places.

**7. Conclusions**

The use of NBS as both a term and as an approach to addressing climate, biodiversity, and health issues within urban planning is growing. Evidence from the Horizon 2020 NBS program highlights the added value of delivering interventions focusing on nature and making direct links between people, location/place, and ecological functions. However, care is needed to ensure that the focus, scale, and location of any individual or program of NBS investment address identified needs and fulfill the climatic, sociocultural, and economic problems associated with a specific location. Moreover, establishing an effective governance regime to co-design, implement, and manage NBS interventions is key to their long-term success. These overarching principles have been partially met from the discussion of the URBAN GreenUP project in Liverpool.

Consequently, the visibility of NBS in Liverpool—and therefore their perceived value to the city—remains variable. Smaller-scale NBS were identified as holding a more limited value due to their size, location, and perceived lack of function, for example, pollinator lamppost investments. Visibly larger interventions and those located within existing NBS, i.e., street trees and water-based NBS, were reported as holding an increased value to residents and professionals. The size and location of NBS within Liverpool were highlighted as critical factors promoting the visibility of each intervention and the subsequent recognition (or lack thereof) of its benefits. Therefore, future investment in NBS would benefit from a more nuanced approach to the choice of interventions with residential and commercial/business communities to ensure greater alignment between strategic and local needs. This was only partially achieved in Liverpool, with significant variations in the reported awareness of NBS. Thus, the lessons to be learned from Liverpool include focusing NBS delivery on a more integrated approach to siting, designing, communicating, and evaluating projects at the local scale. These are valuable lessons applicable to many cities and can act as a valuable learning experiences to aid the effective mainstreaming of NBS as a best practice in city planning.

**Author Contributions:** Conceptualization, I.M., S.C. and F.O.; methodology, I.M., S.C. and F.O.; validation, F.O., I.M. and S.C.; formal analysis, I.M. and S.C.; data curation, F.O.; writing—original draft preparation, I.M. and S.C.; writing—review and editing, I.M. and S.C.; project administration, S.C.; funding acquisition, I.M. and S.C. All authors have read and agreed to the published version of the manuscript.

**Funding:** This project has received funding from the EU's Horizon 2020 Research and Innovation Program under grant agreement number 730426. Support was also received from the University of Manchester through their open access funding agreements.

**Data Availability Statement:** The data used to populate this paper will be made available following the conclusion and validation of the URBAN GreenUP project via EU repositories. Currently, the location of these data has not been made public.

**Acknowledgments:** The authors would like to thank all participants for their considerable thoughts on the URBAN GreenUP project.

**Conflicts of Interest:** The authors declare no conflict of interest.

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
