# Peer review of "Mainstreaming Nature-Based Solutions in City Planning: Examining Scale, Focus, and Visibility as Drivers of Intervention Success in Liverpool, UK"

_land, doi:10.3390/land12071371_

Round 1

Reviewer 1 Report (Previous Reviewer 3)

I appreciate the authors for your continued effort in refining the paper by addressing the comments. Although I have still things to highlight, paper may be accepted with minor changes if there is no issue regarding the contents of paper vis-a-vis scope of journal MDPI Land which may be determined by the Editor. Following are some minor observations:

1. Conclusion needs to be revised thoroughly. There is a need to highlight which type of NBS considered more good compare to others.

2. Abstract also needs revision accordingly.

I have seen in track change mode and may be having some wrong observations.  However, it still needs attention.

Author Response

  1. We thank the reviewer for this comment. We have edited the conclusion and abstract accordingly and made clear that larger scale NBS were considered to be more valuable. Smaller interventions, such as pollinator lampposts, were viewed less favourably due to their size and location.

Reviewer 2 Report (Previous Reviewer 2)

Thanks for the invitation. I found that the manuscript has been improved but some problems still exist. Revisions may be needed to facilitate readers’ understanding and the presentation of the scientific value of this study.

First of all, the document contains some text in red and some text in blue presenting the same information in a sentence. It is almost impossible to know which one the authors want to keep and which one they want to delete. It is thus difficult to review the details of the manuscript.

Then, the biggest problem that remains is the disconnection between the Introduction/Discussion/Conclusion (Sections 1, 2, 3, 6, 7) and the Methods/Results (Sections 4, 5). Sections 1-3 present mainly the URBAN GreenUP project, the case studies in several countries, and different research tasks of the project. But the investigation presented in the manuscript is only one part of the project and focus on the social perceptions of the NBS in Liverpool, UK. The objectives of the project URBAN GreenUP and the objectives of the study presented in Sections 4&5 are not exactly the same. Why do we need the investigation of the social perceptions? What is the scientific value of this investigation for the project and for mainstreaming NBS design in city planning? More need to be clarified in the first three sections. The discussion and conclusions should also be more focused on the results presented in the section 5. They are quite general currently.

In fact, I found the title of the manuscript is clear “Mainstreaming Nature-Based Solutions in city planning: examining issues of scale, focus and visibility as drivers of intervention success”, but the text in the manuscript seems not be able to fully justify or support the title. How are the “issues of scale, focus and visibility” defined? How is the “intervention success” defined? The study is about the perceptions of different stakeholders of NBS. Do social perceptions define intervention success? The authors underline “Elements, functions, benefits, and beneficiaries” (according to their response letter). What are their linkages with the “issues of scale, focus and visibility” mentioned in the title? What are the implications of the study on social perceptions for the consideration of “elements, functions, benefits, and beneficiaries”? The key arguments of the manuscript need to be better articulated and clarified.

Finally, the study investigated the perceptions of different stakeholders. It might be meaningful to present in a clearer manner the implications of different stakeholders’ (business, SME/Social enterprise, NGO/NFP… residents) perceptions for the design and governance of NBS in the Results and Discussion sections because different stakeholders play different roles here.

The writing could be more direct and precise. The sense of some sentences using general or vague terms might be not very clear for readers.

Author Response

Thanks for the invitation. I found that the manuscript has been improved but some problems still exist. Revisions may be needed to facilitate readers’ understanding and the presentation of the scientific value of this study. 

  1. We would like to thank the reviewer for this comment. We are grateful for the comments on the changes and welcome the additional comments on further work needed to ensure that the paper is clearly presented.

First of all, the document contains some text in red and some text in blue presenting the same information in a sentence. It is almost impossible to know which one the authors want to keep and which one they want to delete. It is thus difficult to review the details of the manuscript. 

  1. More than one author edited the paper – hence the different colours of the track changes. We acknowledge that this makes reading difficult in places, as we have also had to contend with the same issue (and being unable to work on a clean version of the paper in order to show the changes). Is it possible double read by clicking on the track change marker on the left of the page. We can provide a fully revised and clean version of the paper on request.

Then, the biggest problem that remains is the disconnection between the Introduction/Discussion/Conclusion (Sections 1, 2, 3, 6, 7) and the Methods/Results (Sections 4, 5). Sections 1-3 present mainly the URBAN GreenUP project, the case studies in several countries, and different research tasks of the project. But the investigation presented in the manuscript is only one part of the project and focus on the social perceptions of the NBS in Liverpool, UK. The objectives of the project URBAN GreenUP and the objectives of the study presented in Sections 4&5 are not exactly the same. Why do we need the investigation of the social perceptions? What is the scientific value of this investigation for the project and for mainstreaming NBS design in city planning? More need to be clarified in the first three sections. The discussion and conclusions should also be more focused on the results presented in the section 5. They are quite general currently.

  1. we thank the reviewer for this comment and the issues they raise. We do not agree with all the comments as we feel that the discussions presented in Section 5 and the outcomes proposed in section 6 provide useful insights into the complexity of investing in NBS and how these investments are received by the stakeholders of a specific city. We have added further commentary in section 3 to highlight the value of the approach taken. We also note that Liverpool is a front runner city in the URBAN GreenUP project and the global investment in NBS. Thus, the findings, especially those related to designing, delivery, and local acceptance of NBS interventions, are important examples of city learning that transfer elsewhere. Understanding resident and professional commentary is therefore useful information that can help cities shape their investment plans. The discussions of scale, focus and visibility (as well as communication), noted in section 6 support this view.

“The following therefore reflects on the perceptions of local respondents to the changes afforded to the physical environment, their interaction and valuing of urban nature, and the socio-economic and ecological benefits that investment in NBS can deliver. This is of value to the city government of Liverpool, and others, as it provided an analysis that reflect on public acceptance of NBS in terms of size, location and type that can be used to shape future investment. It also provides evidence of the role played by communication, engagement and co-design of investment plans for cities, as well as a more nuanced appreciation of how stakeholders make links between alternative environmental and socio-economic factors. Evidence of this nature is valuable to cities in different locations and aids the transferability of best practice (or the identification of poor practice) that other locations can learn from.”     

In fact, I found the title of the manuscript is clear “Mainstreaming Nature-Based Solutions in city planning: examining issues of scale, focus and visibility as drivers of intervention success”, but the text in the manuscript seems not be able to fully justify or support the title. How are the “issues of scale, focus and visibility” defined?

  1. Our definition of scale is focused on the size of NBS interventions and is outlined in the first paragraph (see below) and and in point (3) on the following page. These are also referred to again in the conclusion where smaller-scale interventions are holding less value to respondents compared to larger and more physically visible NBS. Likewise focus is used to report on the functions of NBS, and visibility is the physical visibility, i.e., can respondents se/interact with NBS in situ. This has been added to the text – see below.

“Through a broad program of micro/singular, street and area/neighborhood-based interventions, NBS have been implemented in European cities to test the positive impact that nature-focused interventions can have at multiple scales (micro, e.g., a lamp post, site, e.g., a park or building), street, neighborhood), and across different urban contexts [3]. The following uses scale to define the size of a NBS intervention. In addition, the focus of NBS is reflective of their multiple socio-economic and ecological functions and how they aid the delivery of climate change adaptation/mitigation, health and well-being, economic prosperity, and improved quality of life, whilst the visibility of NBS relates to the ease of which NBS interventions are seen and interacted with in an urban context.”

“Consequently, the visibility of NBS in Liverpool, and therefore, their perceived value to the city remains variable. Smaller-scale NBS were identified as holding a more limited value due to their size, location and a perceived lack of function, for example pollinator lamppost investments. Visibly lager interventions, and those located within existing NBS, i.e., street trees and water-based NBS, were reported as holding an increased value to residents and professionals. The size and location of NBS within Liverpool was highlighted as a critical factor promoting the visibility of each intervention and a subsequent recognition (or lack thereof) of their benefits.”

How is the “intervention success” defined? The study is about the perceptions of different stakeholders of NBS. Do social perceptions define intervention success? The authors underline “Elements, functions, benefits, and beneficiaries” (according to their response letter). What are their linkages with the “issues of scale, focus and visibility” mentioned in the title? What are the implications of the study on social perceptions for the consideration of “elements, functions, benefits, and beneficiaries”? The key arguments of the manuscript need to be better articulated and clarified.

A: We have added additional text in the introductory sections to outline what the paper means by success (see below). We note that success is linked to perceptions of the additional benefits of NBS interventions of respondents. The paper does not propose to provide ecological data showing these changes but relies on commentary to support interpretations of which interventions have been successful. We have also provided additional commentary referring to success at the end of section 5.3 to highlight the variability of what are successful NBS interventions are considered to be.  We also make reference to successful interventions in section 6.4 with reference to the role of communication and physical visibility of NBS as being factors in respondent reporting of successful investment.

 “Within this context successful delivery is framed as NBS investment that are reported by local participants as positively enhancing the quality of place and quality of life socially, economically, or ecologically. Success is not categorized as additional or calculable functionality of urban ecosystems, as this is beyond the scope of this paper. The analysis presented therefore relates to the perceptions of business, environmental and residential stakeholders of the additional benefits that NBS provide in Liverpool.”

Finally, the study investigated the perceptions of different stakeholders. It might be meaningful to present in a clearer manner the implications of different stakeholders’ (business, SME/Social enterprise, NGO/NFP… residents) perceptions for the design and governance of NBS in the Results and Discussion sections because different stakeholders play different roles here.

  1. We thank the reviewer for this comment. We agree with it, but it is beyond the scope of the current paper to go into detail as to how these commentary provided from each group could manifest itself in future practice. We have added in an additional section – new 6.5 – outlining how this may work in practice. This is more of a hypothetical approach rather than stating factually how this could work.

Reviewer 3 Report (Previous Reviewer 1)

I checked the authors' revision. I think my question was well answered.

Author Response

  1. We thank the reviewer for this comment.

This manuscript is a resubmission of an earlier submission. The following is a list of the peer review reports and author responses from that submission.

Round 1

Reviewer 1 Report

This is a very interesting survey report on NBS. Before publishing, I have the following suggestions:

1. This article is less like a scientific paper and more like a report that serves a specific region. I suggest that the authors focus on a scientific question.

2. It is difficult for readers to judge whether these interviews or surveys adequately represent all stakeholders. I suggest adding statistical characteristics of these interviewees in the appendix.

3. I suggest adding some figures and data to illustrate this project instead of just using language descriptions.

4. I think the authors pay more attention to stakeholders' understanding of the project, but they should also consider that some benefits may not be properly recognized by residents.

Reviewer 2 Report

This paper presents the results of a survey in the Horizon 2020 URBAN GreenUP project of the social perceptions for Nature-Based Solutions projects in Liverpool, UK. The topic is interesting, but I found the writing a little fragmented. I would like to suggest minor revisions about the following points to reinforce its clarity:

1. I think the title could be more precise by adding “Nature-Based Solutions in city planning” or “in urban planning”, like in the text.  

2. There seems to be a disconnection between the conclusions highlighted in the Abstract and the supporting evidence in the Results. For example, L22-23 “a more nuanced appreciation of the cumulative benefits linked to networks of green and blue spaces may offer greater visibility”, I am not sure that the survey results leading to this conclusion have been clearly presented in Section 3 (which should be named as Section 5 after “4. Materials and Methods”). I also feel the disconnection between the presentation about the Methods, Results, and Discussion. It would help to improve the clarity of the manuscript by following the same main points in each section.

3. The Introduction is more like a Conclusion. It would be better to clarify and strengthen the arguments which justify the research objectives. For example, L38 “considerations of scale, location, NBS function, visibility, and interactivity”, what does each of these terms mean for Nature-based Solutions? The text goes too fast here.

4. This paragraph appears repeatedly: “the model developed by Kabisch, Frantzeskaki & Hansen [1] who proposed that effective NBS investment be structured against five core principles: (a) systematic understanding, (b) benefits to people and biodiversity, (c) inclusive solutions that are long-term, (d) context consideration, and (e) communication and learning”, in Introduction (L56-60), Section 2 (L133-137), and Discussion (606-609). It gives the impression that this is the fundamental departing point and conclusion of this study, but the research methods and results did not follow the structure (a)-(e) here. What is the linkage between the authors’ survey and the model of Kabisch et al.? There might be a better way to introduce the value of this model for the authors’ study.

Reviewer 3 Report

I have examined the manuscript thoroughly and also awaited copy of questionnaire and data file so as to provide a good review particularly by scrutinizing the requested supplementary material upon which the paper can be improved. Some aspects related to qualitative and quantitative data analysis were not clear. Unfortunately, despite my repeated request to Land Editorial Team, I could not get the desired copies of questionnaire and data file from authors (through MDPI Land Office). Whereas, it was also not made available online in Land Submission System. Although the authors have chosen a good, I have following observations:

1.     The quality of research and material presented don’t suffice to qualify for publication in MDPI Land.

2.     From heading # 1,2 & 3, the contextual framework for the paper vis-à-vis the desired objective, scope of work and the title is not well articulated. There is no NBS mentioned against which the perception has been studied. There is a need to revisit these sections thoroughly and mention NBS against which the design of study should also be properly revisited.

3.     The current design of the study is not coherent vis-à-vis problem statement, title, objective and scope. Methodology is underdeveloped as the NBS are neither identified nor described upon which the variable could have been fixed to examine the perception as was desired by the authors. This creates a big contradiction between the contents, intention of the authors and the title of the paper, which is also an indicative of a disjointed effort made by the authors in preparing this manuscript.

4.     The data collection is reported through 75 ex-anti and 173 post-anti (vide lines 279-283) survey based responses on questionnaire; and a total of twenty-two semi-structured interviews reportedly (vide lines 232-233) conducted with businesses, SMEs and social enterprises, non-governmental organizations workers, members of ‘Friends of’ groups, and elected officials/councilors. However, parameters / variables or contents / scope for both type of questionnaires is not clear due to which it is very difficult to infer anything. Besides, although the authors have used word ‘NBS’ frequently at times and again, the type or category or names of NBS are not described due to which the basic research design has a big question mark.

5.     Likert Scale is mentioned for quantitative data collection, however, it is not clear how output of twenty two semi-structured interviews is clubbed or integrated with quantitative part? In Table 2, results reported against six questions become out of context when the compared with the title. Title of manuscript mentions interventions which are categorically linked with the type of NBS. However, actual NBS are not identified as already highlighted above. Results in Table 3 against 10 queries are also not well integrated with the actual study design. Whereas, it is had to understand what the authors want to convey through Table 4.

6.     During reading the manuscript, punctuations and other grammatical errors are frequently observed at times and again e.g. in line number 101 “support an more holistic approach”. The entire Manuscript needs a thorough English editing.

7.     There is no new knowledge addition from this paper.

8.     Considering above key comments and many other issues in the manuscript, it is advised to re-write and re-submit the paper. Hence, I hereby recommend to decline the paper as the entire paper needs re-writing and re-submission.